

# Size and body condition drive the energetic cost of a baleen whale foraging in shallow habitat

Clara N. Bird[1], Enrico Pirotta[2], Leslie New[3], Jamie M. Cornelius[4], James L. Sumich[5], Kate M. Colson[1,6], K.C. Bierlich[1], Lisa Hildebrand[1], Alejandro Apolo Fernández Ajó[1], Annie Doron[1,7] and Leigh G. Torres[1]

[1] Geospatial Ecology of Marine Megafauna Lab, Marine Mammal Institute, Department of Fisheries, Wildlife and Conservation Sciences, Oregon State University, Newport, OR, United States of America

[2] Centre for Research into Ecological and Environmental Modelling, University of St Andrews, St Andrews, United Kingdom

[3] Ursinus College, Collegeville, PA, United States of America

[4] Integrative Biology, Oregon State University, Corvallis, OR, United States of America

[5] Marine Mammal Institute, Department of Fisheries, Wildlife and Conservation Sciences, Oregon Statec University, Corvallis, OR, United States of America

[6] Marine Mammal Research Unit, Institute for the Oceans and Fisheries, University of British Columbia, Vancouver, British Columbia, Canada

[7] University of Sheffield, Sheffield, United Kingdom

Corresponding author
Clara N. Bird,
clara.bird@oregonstate.edu,
clara.birdferrer@gmail.com

## ABSTRACT

Energy expenditure strongly influences an animal's foraging decisions and activity budgets. Diving animals especially need to be energetically efficient because they exercise while oxygen is limited. By estimating the energetics of behavior, we can better understand the cascading effects of individual responses to disturbance and environmental change. Pacific Coast Feeding Group (PCFG) gray whales use a variety of foraging tactics in shallow habitats (<20 m), which present challenges associated with maneuverability and buoyancy. We use a seven-year dataset of concurrent individual behavior, morphology, and breath-by-breath respiration data collected via drone paired with two years of tri-axial accelerometry tag data to study patterns and correlates of respiration. We assess how several respiration metrics (acting as proxies for oxygen consumption) are associated with individual length, body condition and behavior (forage and travel), and test whether respiration reflects recovery from, or anticipation of, a foraging dive using Bayesian linear mixed effects models. Given model results, we simulated daily field metabolic rate (FMR) to explore how diving costs may affect energetics at a daily scale. We find that respiration reflects recovery from the preceding dive and that dives are more energetically expensive for longer, more buoyant whales. Longer dives and the most common foraging tactics also incur higher energetic costs. FMR simulations show that individual size and dive duration have the largest effects on energy expenditure. Thus, PCFG gray whale foraging success may be limited by the energetic costs associated with size and buoyancy, highlighting the costs of a shallow habitat foraging niche.

## INTRODUCTION

An individual's activity and energetic budgets are fundamentally linked, as locomotion requires energy. Measuring both behavior and energy expenditure in tandem can inform the causes and consequences of an individual's decisions, such as resource or space use (*Brown et al., 2004*). In the context of foraging, when animals are both gaining and spending energy, the profitability of different prey types depends on their energetic value and the energetic cost of pursuing, capturing, and handling the prey (*Emlen, 1966*; *Schoener, 1971*). In turn, understanding foraging energetics and decision-making can support conservation efforts, particularly through studies of how behavioral responses to disturbance could scale to population-level impact (*Cooke et al., 2014*; *Pirotta et al., 2018*). However, measuring energy expenditure in wild animals can be challenging and often involves proxy measures of metabolic rate.

Air-breathing, diving animals, including cetaceans, are among the most challenging animals to study behavior and energy expenditure, as they often expend most energy during dives while submerged and oxygen limited (*Irving, 1939*). Given the need for oxygen, the ocean's surface can be considered a central place and dives can be studied within the framework of central place foraging theory (*Doniol-Valcroze et al., 2011*); animals have a limited amount of time before they must return the surface. Consequently, efficient use of oxygen primarily determines the accessibility and profitability of prey (*Houston & McNamara, 1985*). Accordingly, diving animals have several adaptations to efficiently store and use oxygen during apneic periods (*Scholander, 1940*; *Kooyman & Ponganis, 1998*).

Oxygen stores vary between species due to differences in body size, dive behavior, and physiology (*Scholander, 1940*; *Cartwright et al., 2016*). Both oxygen stores (*Kooyman, 1973*; *Noren & Williams, 2000*; *Sumich & May, 2009*) and consumption (*Kleiber, 1975*; *He et al., 2023*) allometrically scale with body size, though by different factors. The energetic cost of a dive also depends on more dynamic variables, as differing levels of exercise associated with activity state and buoyancy affect oxygen use (*Williams et al., 2015*; *Ponganis & Williams, 2016*).

Oxygen acquisition in between consecutive dives has been associated with both recovery from and anticipation for dives (*Ridgway, Scronce & Kanwisher, 1969*; *Isojunno et al., 2018*). Measures of surface duration, heart rate, breath count, oxygen acquisition, and respiration rate during recovery surfacings indicate that longer dives cost more energy (*Kooyman, 1973*; *Dolphin, 1987a*; *Keen & Qualls, 2018*), though this energetic cost also varies across behavior states and travel speeds (*Butler & Woakes, 1979*; *Sumich, 1983*; *Dolphin, 1987b*; *McRae et al., 2024*; *Spina et al., 2024*). Respiration patterns associated with anticipation for a dive include hyperventilation (*Ridgway, Scronce & Kanwisher, 1969*; *Dolphin, 1987a*) and longer terminal breath durations (*Nazario et al., 2022*). Respiration at the surface can also reflect recovery from oxygen-debt accumulation or delayed carbon dioxide removal across multiple dives (*Fahlman et al., 2008a*; *Génin et al., 2015*). Deeper dives are generally thought to be more energetically expensive as they tend be longer than shallow dives (*Doniol-Valcroze et al., 2011*), but deep divers can use buoyancy to their advantage by gliding to reduce energetic costs (*Williams, 1999*). In contrast, buoyancy only increases

the cost of shallow dives (*Kooyman, 1973*; *Stephenson et al., 1989*; *Lovvorn & Jones, 1991*). Therefore, while the dive physiology of deep divers is well studied, the adaptations and costs associated with shallow divers are less understood.

Respiration metrics can serve as accessible proxies for oxygen consumption (*i.e.,* energetic expenditure; *Scholander, 1940*). While respirometry provides the most direct measures of oxygen consumption, it requires handling (*Olsen, Hale & Elsner, 1969*; *Sumich, 2001*; *Fahlman et al., 2008a*), which is infeasible, impractical, or unethical for most cetaceans. Surface ventilation patterns of wild cetaceans can be measured non-invasively (*e.g.,* respiration rate, inter-breath interval, surface duration; *Sumich, 1983*; *Dolphin, 1987a*; *Dolphin, 1987b*; *Stelle, Megill & Kinzel, 2008*; *Williams & Noren, 2009*; *Christiansen, Rasmussen & Lusseau, 2014*; *Keen & Qualls, 2018*). However, converting these remotely measured variables into consumed oxygen relies on assumptions around the tidal volume (*i.e.,* the amount of air exchanged with each respiratory cycle) and oxygen extraction efficiencies (*Fahlman et al., 2016*), which can introduce high uncertainty (*Winship, Trites & Rosen, 2002*).

Relative to all other baleen whales, gray whales (*Eschrichtius robustus*) have a rich history of respiration studies, making them an excellent species for estimating energetic expenditure based on respiration metrics. Respirometry studies on gray whale calves, both restrained and in human care, have quantified oxygen extraction efficiency, flow rate, inhalation and exhalation durations, and tidal volume (*Wahrenbrock et al., 1974*; *Kooyman, Norris & Gentry, 1975*; *Sumich, 2001*). These studies found that, in a single breath, the inhalation and exhalation durations were equal, flow rate peaked halfway through an inhalation or exhalation, oxygen extraction efficiency increased with inhalation duration, and tidal volume can be accurately estimated from body length and inhalation duration (*Sumich, 2001*; *Sumich & May, 2009*). While gray whale respiration rates in the field have been described during migration (*Sumich, 1983*) and foraging (*Wursig, Wells & Croll, 1986*; *Mallonee, 1991*; *Stelle, Megill & Kinzel, 2008*), these studies did not link respiration metrics to morphology or activity levels.

Gray whales are unique within baleen whales because they are suction feeders (*Nerini, 1984*). Within this species, Pacific Coast Feeding Group (PCFG) gray whales, which comprise a small (~212 individuals; *Harris et al., 2022*) subgroup of the Eastern North Pacific (ENP) population of gray whales (~19,260; *Eguchi, Lang & Weller, 2024*), feed in shallow, coastal habitats (<20 m depth; *Bird et al., 2024a*) between Northern California, USA, and Southern British Columbia, Canada (*Calambokidis, Perez & Laake, 2019*) during the summer months (June–November). PCFG gray whales use at least eight different foraging tactics (*Torres et al., 2018*; *Bird et al., 2024a*) to forage on a variety of benthic, epibenthic and pelagic invertebrates (*Darling, Keogh & Steeves, 1998*; *Dunham & Duffus, 2001*; *Hildebrand, Bernard & Torres, 2021*) in both rocky reef and sandy habitats (*Bird et al., 2024a*). These large capital-breeding baleen whales forage in a shallow niche despite having no obvious morphological adaptations for shallow diving (*e.g.,* unlike sirenians' dense bones, horizontal diaphragm and skeletal weight distribution; *Domning & de Buffrénil, 1991*). Despite this apparent lack of adaptation, PCFG gray whales persistently feed in this shallow niche and show high maternal recruitment (*Lang et al., 2014*; *Calambokidis & Pérez,*

*2017*), suggesting that this strategy is beneficial; however, a comprehensive assessment of their energetics during these shallow foraging dives is needed to better understand their ecology, physiology and response to disturbance.

There is individual variation in the use of foraging tactics across PCFG gray whales, and an ontogenetic shift in tactic use associated with growth in length (*Bird et al., 2024a*). Shorter, younger whales utilize forward moving behaviors and longer, older whales predominantly use stationary behaviors (*e.g.*, headstands; *Bird et al., 2024a*). Despite being one of the most prevalent tactics, headstanding appears to be costly. Fluke stroke rate, derived from accelerometry tags deployed on PCFG gray whales (*Colson et al., 2024*), indicates that headstanding is the most energetically costly foraging tactic relative to other tactics (*Colson, 2023*). However, PCFG gray whales appear to use bubble blasts, underwater releases of air that rises to surface and forms a circle/puka (*Torres et al., 2018*), as a behavioral adaptation during foraging dives to reduce the costs of buoyancy and extend dive times (*Bird et al., 2024b*). Combined, the links between body size, shifts in preferred foraging tactic, and bubble blast occurrence suggest that PCFG gray whales have developed behavioral adaptations to forage in this shallow niche, but energetic costs are variable by tactic, individual, and habitat type.

Drones and accelerometry tags can provide valuable respiration data, offering the potential to pair energy estimates with behavior. High-resolution, breath-by-breath metrics can be extracted from video footage (*Nazario et al., 2022*; *Sumich et al., 2023*), while breath counts and activity can be measured from tag data using accelerometry, depth, or acoustic data (*Fahlman et al., 2008b*; *Halsey et al., 2008*; *Roos, Wu & Miller, 2016*; *Isojunno et al., 2018*). Drones also provide concurrent individual morphology and body condition data (*Dawson et al., 2017*; *Torres et al., 2022*; *Bierlich et al., 2023*). Therefore, the high-resolution respiration data derived from these tools can advance our understanding of diving energetics in baleen whales (*Mysticeti*), for which physiological information is limited relative to other marine mammals (*Fahlman et al., 2016*). Furthermore, the shallow coastal habitat and site fidelity of PCFG gray whales provides an ideal system in which to study baleen whale energetics with high resolution data as individuals are accessible to be repeatedly sampled over time.

Here, we explore how morphology, body condition, and behavior affect PCFG gray whale respiration in a shallow foraging habitat. We aim to measure how energetic costs change with different dive characteristics and hypothesize that the energetic cost of a foraging dive increases with individual body length, body condition (associated with increased buoyancy) and dive duration and varies across different foraging tactics. Specifically, we use a unique seven-year longitudinal dataset of PCFG gray whale morphology, body condition, and behavior from drone footage and a two-year dataset of tag deployments to first examine metrics that could be used to test our hypothesis by (1) exploring a suite of respiration metrics that reflect PCFG gray whale energetics on their foraging grounds and (2) comparing respiration metrics derived from drone and tag data. We then test our hypothesis by (3) modeling how respiration metrics vary with body length, body condition, and behavior, (4) discerning whether respiration metrics indicate recovery from

or anticipation of a dive, and (5) simulating daily FMR using our results to contextualize energetic demands on a daily scale.

## MATERIALS & METHODS

### Data collection

We collected drone footage of PCFG gray whales filmed off the coast of Newport (44.60765, −124.08162) and Port Orford (42.737407, −124.505301), Oregon, United States, between June and October of 2016–2022 (Fig. 1). Drone operations were conducted in good weather conditions from a small (5.4 m) rigid hull inflatable boat (wind <22 km/h, swell <1.5 m, minimal fog or rain). *Ad libitum* (*Altmann, 1974*) boat-based surveys were conducted by teams of 3–4 people, and included photo-identification of individual whales.

Four different DJI drones were used over the course of this study: a Phantom 3 Pro, 4 Advanced and 4 Pro, and an Inspire 2. A laser altimeter (*e.g.*, "LidarBoX") was mounted on the Inspire 2 to provide more accurate photogrammetry measurements (*Dawson et al., 2017*; *Bierlich et al., 2024*). Details on the cameras mounted on each drone are available in supplementary (Table S2). The pilot recorded video continuously during flight (~15 min) at an altitude between 20 and 40 m, during which the whale was located and followed while visible at the surface or underwater.

Throughout the 2021 and 2022 field seasons, nine suction-cup-attached Custom Animal Tracking Solution (CATS; https://cats.is) video and inertial measurement unit tags were deployed on nine individual whales using an 8 m carbon fiber pole from the boat (*Colson et al., 2024*). Tags were equipped with a gyroscope, 50 Hz magnetometer, 400 Hz accelerometer, 10 Hz pressure, temperature, light and GPS sensors, and a video camera and hydrophone. Each of the four suction cups on the tag had an oxidizing release to guarantee release after suction failed. All nine individuals tagged were also in the drone dataset.

Data was collected using drones and suction cup tags. Research was conducted under NOAA/ NMFS permits #16011 and #21678 and IACUC approval No.CRC-SOP-3. UAS operations were conducted by a Federal Aviation Authority (FAA) certified private pilot with a Part107 license. During field work, behaviour was observed for indications of a behavioural response to disturbance (changes in direction, increased swim speed, sudden dives) and none were observed. Disturbance potential was minimized by maintaining a slow boat speed when near whales and maintaining the drone at a minimum altitude of 20 m.

### Data processing
#### Drone data

*Video processing.* Drone footage was first clipped to periods when a whale was visible, and the whale(s) in each clip were identified using an aerial catalog and photo-identification images taken during the flight. A single experienced analyst (CNB) then reviewed each clip at least twice per whale to annotate the behaviors performed using continuous focal sampling and the Behavioral Observation Research Interactive Software (BORIS; *Friard & Gamba, 2016*). We used an ethogram containing 49 behaviors (Table S1, *Torres et al., 2018*; *Bird et al., 2024a*). For this study we focused on forage and travel behaviors (Table 1). Foraging

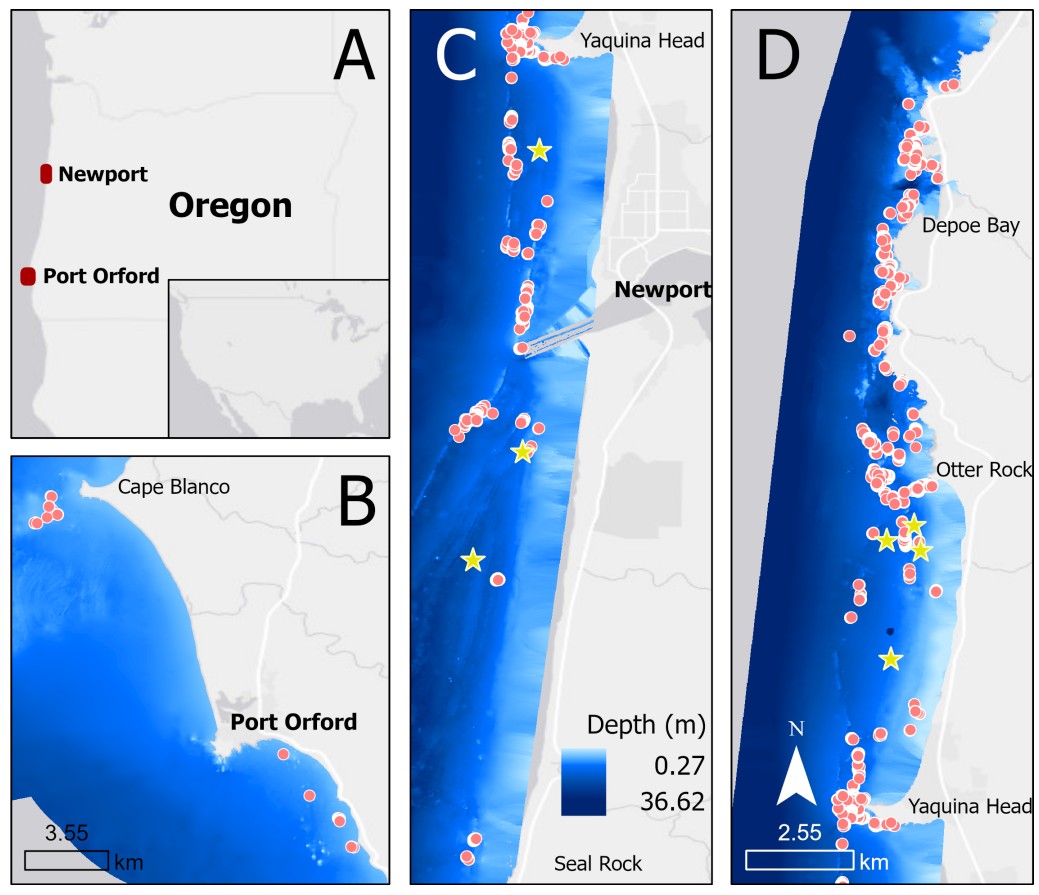

**Figure 1** Map showing locations of tag deployments (yellow stars) and drone observations (red dots) in Oregon (A), specifically off Port Orford (B) and Newport (split into C and D).

behavior was defined down to the tactic level. Foraging tactics that were mechanically similar and tended to co-occur were grouped to increase sample size following *Bird et al. (2024a)* (Table 1). Therefore, all behaviors exhibited during a dive sequence were assignable to a single, mutually-exclusive category—except for bubble blasts, which could occur in coincidence with foraging tactics. Further filtering was applied to select only clips with at least one complete surfacing sequence in the footage, which was defined as sequences where the whale surfaced for the first time after a dive and started a new dive at the end of the sequence. Within each complete sequence, breaths were categorized as initial (the first breath in the sequence), terminal (the final breath in the sequence), or middle (any breath between the initial and terminal breaths). In single breath sequences, the breath was classified as initial.

*Respiration metrics.* We quantified the following respiration metrics from drone video review: surface sequence duration, individual breath inhalation duration, total inhalation duration, inter-breath interval (IBI), rate of inhalation accumulation, breath count, respiration rate, and nares area. Surface sequence duration was calculated as the number

**Table 1 Ethogram containing definitions of PCFG gray whale behaviors included in this study.** The complete ethogram is available in Table S1.

| Primary behavior state | Tactic group | Sub-behavior tactics | Definition |
|---|---|---|---|
| | Headstand | Headstand | Whale is positioned head down-flukes up, or if in water depths less than whale body length, whale may be more horizontal in water column; With both body positions the whale is observed pushing head/mouth region into substrate. |
| | Side-swim (stationary) | Side-swim (stationary) | Whale observed swimming on its side, but not moving forward. Characterized by frequent jaw snapping. |
| | Forward swimming tactics | Side-swim (forward) | Whale observed swimming on its side, moving forward. Characterized by frequent jaw snapping. |
| | | Upside-down swim (forward) | Whale observed swimming upside-down, moving forward. Characterized by frequent jaw snapping. |
| | | Subsurface (forward) | Whale swims subsurface while feeding. Characterized by frequent jaw snapping. |
| | Subsurface (stationary) | Subsurface (stationary) | Whale maintains a stationary position while feeding below the surface of the water, oriented dorsal up. |
| Forage | Surface tactics | Surface feeding | Whale feeding at the surface, frequently breaking the surface but without breathing. Characterized by frequent turning and frequent jaw snapping/flexing. |
| | | Skim feeding | Whale swims at the surface with mouth open for an extended period. Characterized by moving forward in a straight line. |
| | Bubble blast | Bubble blast | Underwater release of air by whale that rises to surface and forms a circle/puka. |
| Travel | | | Whale shows directed travel in a consistent direction, with regular surfacing intervals. |

of seconds between the initial and terminal breaths. To calculate the inhalation duration of each breath, an analyst noted the start and end time of each breath (including exhalation immediately followed by an inhalation). The start was denoted as the time when bubbles started flowing out of the nares and the end as the moment the nares closed or became submerged underwater. Inhalation duration was calculated as one-half of the breath duration (*Sumich et al., 2023*). Total inhalation duration for a surface sequence was the sum of all the individual breath inhalation durations. IBI was the mean number of seconds between sequential breaths in a surface sequence (*Sumich, 1983*; *Stelle, Megill & Kinzel, 2008*). The rate of inhalation accumulation was calculated as the slope of the relationship between total inhalation duration and seconds since first breath, which represents a combination of inhalation duration and respiration rate and may reflect recovery strategy (*i.e.,* fast *versus* slow). One slope was calculated per surface sequence. The breath count was the total number of breaths observed in the surface sequence. Respiration rate (breaths min$^{-1}$) was calculated for travelling whales as the total breath count for the observation period divided by the total travel duration for that whale (*Sumich, 1983*; *Spina et al., 2024*).

To explore the relationship between nares area and oxygen exchange we measured the maximum nares area (m$^2$) during an inhalation. To measure maximum area, a snapshot was extracted at the midpoint of the inhalation when the flow rate was highest (*Sumich & May, 2009*). Snapshots were then filtered for quality; only clear images where the nares were

not obstructed by the blow were retained (Fig. S1, Table S3). Nares were measured using MorphoMetriX (*Torres & Bierlich, 2020*) and processed using CollatriX (*Bird & Bierlich, 2020*). The length and width were measured using the 'Measure Length' function and area was measured using 'Measure Area' (Fig. S2). In a subset of 30 randomly selected snapshots, both the left and right nares were measured to test the assumption that they were equal. In all remaining images, only the clearest naris was measured. For comparisons across individuals, the area was standardized by the individuals' total length (TL). This metric was only measured on a subset of the data (2020–2021; $n = 183$ breaths).

*Individual length and body condition.* The total length (TL) and body area index (BAI) of each individual in this study was estimated from snapshots extracted from drone footage following photogrammetry methods described in *Torres et al. (2022)*. Measurements were made manually using MorphoMetriX (*Torres & Bierlich, 2020*) and processed using CollatriX (*Bird & Bierlich, 2020*). A posterior predictive distribution for TL was generated using a Bayesian statistical model to account for photogrammetric uncertainty and growth (*Bierlich et al., 2021b*; *Pirotta et al., 2024*). One TL per individual per year was estimated, as individuals were not expected to significantly grow during a foraging season. BAI, a standardized measure of body condition that changes more rapidly than TL (*Burnett et al., 2018*; *Bierlich et al., 2021a*), was calculated on the same day for which behaviors were scored. If there were no BAI measurements from the date of the behavior observation, the nearest measurement within ±14 days was selected (*Pirotta et al., 2023*).

*Travel swim speed.* Swim speed during travel was derived from the drone's GPS (*Christiansen et al., 2023*). Speed was only calculated for periods when the drone was directly over the whale. GPS position was extracted every 5 s and the distance between the points was divided by time to calculate speed in m s$^{-1}$. Speed was then averaged across the entire observation period.

### Tag data

We used data from nine tag deployments on nine individual whales. The mean deployment duration was 9.14 hr (range = 0.44 h to 24.81 hr; *Colson et al., 2024*).

*Behavior.* We used the primary behavior state (forage, search, and travel) and foraging tactic (headstand, side swim, and benthic dig) event classifications from *Colson et al. (2024)*. *Colson et al. (2024)* assigned primary behavior states based on turn angle, dive duration, pseudotrack tortuosity, and roll presence using hidden Markov models, while foraging tactics were classified by median pitch, absolute median roll, and the depth to TL ratio using a classification and regression tree (CART) model (with 84.8% accuracy). Consecutive travel dives were grouped into a travel series.

*Breath count.* Surface sequences between dives were identified during manual examination of the tag deployment's depth profile (*Colson et al., 2024*). Breaths during a surface sequence were identified as surfacings that occurred between very shallow dives identified using the 'find_dives' function from the package *tagtools* (*DeRuiter et al., 2024*) with a minimum
depth of 0.2 m. To validate our methods, we compared breath timestamps identified from tag and drone data during a single six-minute clip when the drone was over a tagged whale; this validation confirmed that the surfacings between shallow dives aligned with breaths in the drone footage. We also determined that, in the tag data, any surfacing longer than 10 s represented at least two breaths. Thus, we counted every surfacing longer than 10 s as two breaths. This breath count metric represents a minimum estimate, given that breaths could be missed. Respiration rate was calculated for traveling whales as the minimum breath count divided by the total travel series time.

## Statistical analysis

Analyses were performed in R v4.4.2 (*R Core Team, 2024*). Bayesian models used *RStan* (*Stan Development Team, 2020*), and *rethinking* (*McElreath, 2020*). All data and code are available at: https://doi.org/10.6084/m9.figshare.26999995.v1.

### Metric exploration

*Nares area.* Frequentist linear models were used to assess the relationships between the right and left nares (fitting separate models for length, width, and area), and the relationships between TL and the mean nares length, width, and area. Finally, a linear mixed effect model (LMM) was used to assess how nares area, standardized by TL, varied across breath types (initial, middle, terminal), accounting for individual ID as a random effect. Nakagawa's $R^2$ was used to assess the goodness-of-fit of the LMMs (*Nakagawa & Schielzeth, 2013*).

*Correlation of metrics.* Prior to model development, all respiration metrics from the drone data were summarized at the surface sequence scale. Pearson correlation coefficients were used to explore linear correlations between the respiration metrics.

*Respiration models.* Several models were run to investigate how each respiration metric varied as a function of body length, body condition, and behavior (Table 2). We ran separate foraging and travelling models; in the foraging models behavior included bubble blast occurrence, dive duration, and tactic, and in the travelling models behavior was only swim speed. To incorporate uncertainty in the measurement of TL and BAI, their values were imputed within the Bayesian model from their posterior distributions. TL, BAI, mean swim speed, and dive duration were z-score standardized and used as fixed effects. Continuous respiration metrics were log-transformed to meet model assumptions. All models were fit in a Bayesian framework as it allows for the incorporation of photogrammetric uncertainty associated with drone-based measurements. Prior distributions for all model parameters are given in Tables S8, S10, S11, S14 of the supplementary. For all models, we assessed convergence using effective sample size, $\hat{R}$ values, and visual examination of trace plots (*McElreath, 2020*). Visual examination of the residuals, bayes $R^2$ (*Gelman et al., 2019*), and posterior predictive checks were used to assess model fit. Model coefficients were interpreted using the posterior 95% credible intervals and their percent overlap with zero.

Bird et al. (2025), *PeerJ*, DOI 10.7717/peerj.20247

**Table 2  All models and metrics.**

| Hypothesis | Response[a] | Fixed effects[b] | Timing | Random effect | Error structure |
|---|---|---|---|---|---|
| Foraging: Recovery | Breath count (drone) | TL, BAI, dive duration, bubble blast occurrence, foraging tactic | | Individual ID | Poisson (log link) |
| | Breath count (tag) | TL, BAI, dive duration, foraging tactic | | Individual ID | Poisson (log link) |
| | Total inhalation duration | TL, BAI, dive duration, bubble blast occurrence, foraging tactic | | Individual ID | Gaussian (log transformed response) |
| | Mean inter-breath interval (IBI) | TL, BAI, dive duration, bubble blast occurrence, foraging tactic | Bubble blast occurrence, dive duration, and foraging tactic from the dive following the surface sequence | Individual ID | Gaussian (log transformed response) |
| | Rate of inhalation accumulation | TL, BAI, dive duration, bubble blast occurrence, foraging tactic | | Individual ID | Gaussian (log transformed response) |
| | Intial breath inhalation duration | TL, BAI, dive duration, bubble blast occurrence, foraging tactic | | Individual ID | Gaussian (log transformed response) |
| Foraging: anticipation | Total Inhalation Duration | TL, BAI, dive duration, bubble blast occurrence, foraging tactic | | Individual ID | Gaussian (log transformed response) |
| | Mean inter-breath interval (IBI) | TL, BAI, dive duration, bubble blast occurrence, foraging tactic | Bubble blast occurrence, dive duration, and foraging tactic from the dive following the surface sequence | Individual ID | Gaussian (log transformed response) |
| | Rate of inhalation accumulation | TL, BAI, dive duration, bubble blast occurrence, foraging tactic | | Individual ID | Gaussian (log transformed response) |
| | Terminal breath inhalation duration | TL, BAI, dive duration, bubble blast occurrence, foraging tactic | | Individual ID | Gaussian (log transformed response) |
| Foraging: recovery and anticipation | Total Inhalation Duration | TL, BAI, dive duration, foraging tactic | | Individual ID | Gaussian (log transformed response) |
| | Mean inter-breath interval (IBI) | TL, BAI, dive duration, foraging tactic | Dataset was filtered to only surface sequences where the same foraging tactic was used in both the preceding and following dive | Individual ID | Gaussian (log transformed response) |
| | Rate of inhalation accumulation | TL, BAI, dive duration, foraging tactic | | Individual ID | Gaussian (log transformed response) |
| Travel | Respiration rate | TL, BAI, travel swim speed | | Individual ID | Gaussian (log transformed response) |

**Notes.**
[a]All response metrics were log transformed.
[b]All fixed effects were scaled.

*Foraging models.* We checked for autocorrelation in model residuals using Poisson log-linear models for breath count. We selected this respiration metric as it was available from both drones and tags and was highly correlated with total inhalation duration (Fig. S6). The model for breath counts from the drone data included TL, BAI, preceding dive duration, bubble blast occurrence, and foraging tactic as fixed effects. The model for breath counts from the tag data included the same fixed effects except for bubble blast occurrence, which was unavailable from the tag data.

LMMs were used to assess how the natural log of total inhalation duration, IBI, the rate of inhalation accumulation, breath count, and the initial or terminal breath inhalation durations varied with TL, BAI, and behavior. We ran models to test the hypotheses that respiration is informative of a whale's recovery from a dive, their anticipation of the next dive, or some combination of both. The recovery models used the bubble blast occurrence, dive duration, and foraging tactic from the dive preceding the surface sequence. The anticipation models used behaviors from the dive following the surface sequence. The combined model used only surface sequences where the preceding and following foraging tactics were the same, did not include bubble blast occurrence as a fixed effect, and included both the preceding and following dive durations as fixed effects. TL and BAI were included in all models along with an individual-level random effect. The other explanatory variables differed amongst the model sets depending on data availability (see Table 2). This approach gave a total of 12 models, each of which was run for three chains of 90,000 iterations with the first 30,000 as warm-up.

*Travel model.* We used an LMM to evaluate the relationship of respiration rate during travel (derived from the drone data) with TL, BAI, and mean swim speed, including individual ID as a random effect. We ran three chains of 30,000 iterations with 10,000 as warm-up.

### FMR and prey requirement simulation

To contextualize model results within gray whale energetics, we used Monte Carlo simulations to estimate field metabolic rate (FMR) in megajoules (MJ) at a daily scale using respiration metric values predicted from the models described above. For the simulations, we investigated the effects of TL, BAI, dive duration, and behavior on FMR. We simulated two TL values (9 and 12 m), three BAI values (22, 27, 32), three dive durations (60, 120, 300 s), and two foraging tactics (headstanding and forward swimming), thus capturing the range of explanatory variables evaluated for PCFG whales. These values resulted in 36 unique combinations of covariates for which FMR was simulated. Before running the simulations, we used the recovery models to predict total inhalation duration, breath count, and IBI for each of the 36 unique covariate combinations.

Daily FMR was calculated following the methods described in *Villegas-Amtmann et al.* *(2015)*:

$$FMR_{daily} = H \cdot \%O_2 \cdot V_{T(daily)}.$$  (1)

For heat production, $H$ (MJ L$^{-1}$ O$_2$ consumed), we used a value of 0.002 from *Kleiber* *(1961)*. For O$_2$ extraction efficiency, $\%O_2$, we drew a value from a normal distribution

at each iteration, using the mean (11%) and standard deviation (2.7%) calculated by *Villegas-Amtmann et al. (2015)* using data from *Sumich (2001)*. We estimated tidal volume ($V_T$) following Eq. (2) (*Sumich et al., 2023*), where tidal volume is estimated using the inhalation (or exhalation) duration ($t_{in}$) of a breath multiplied by the square of TL. While most studies multiply the tidal volume for a single breath by an estimated respiration rate to calculate FMR, we calculated a total daily tidal volume by multiplying the square of TL by the sum of the total inhalation durations within a day, $t_{in(daily)}$. As a result, daily tidal volume was:

$$V_{T(daily)} = -7.24 + 2.14 \cdot t_{in(daily)} \cdot TL^2. \tag{2}$$

To calculate the sum of the total inhalation durations within a day, we first divided the day into foraging hours and travel/search hours using the activity budget from *Colson et al. (2024)* (36% of time foraging, 43% of time searching, and 21% of time traveling). For the purposes of this simulation, travel and search were combined as their stroke rates are not significantly different (*Colson, 2023*).

For each of the 36 covariate combinations, we simulated complete dive and recovery cycles during foraging periods. We began by drawing a value for the breath count from the corresponding posterior distribution from the recovery models. This value was used as the mean of a Poisson distribution to generate a simulated breath count for each surface sequence. The simulated breath count determined the number of IBIs that followed a dive, and the length of these intervals was drawn from the posterior distribution of IBI from the recovery models. Total inhalation duration was also drawn from the corresponding posterior distribution. Together, the simulated dive duration, summed IBIs, and total inhalation duration represented a complete dive and recovery cycle. We simulated complete dive and recovery cycles for the foraging period until the total time spent in this activity state was equal to the expected number of hours in a 24-hour period. The same process was carried out for the travel/search hours; however, we resampled breath hold and inhalation durations from distributions with mean and standard deviation derived from the entire travel dataset, as our travel models showed no notable relationships between respiration rate and TL, BAI, or speed (see 'Results' section). Within the travel/search period, we iteratively drew a breath hold duration from a lognormal distribution of breath holds (log mean: 3.66, s.d: 0.65) and an inhalation duration from a lognormal distribution of inhalation durations (log mean: 0.26, s.d: 0.17). The two values were then added, and the process was repeated until the sum reached the total travel/search time for the day.

The total inhalation durations from both foraging and travel/search hours were summed to obtain $t_{in(daily)}$. Total daily tidal volume (Eq. (2)) was then used to calculate daily FMR (Eq. (1)). Finally, within each simulation, we estimated prey requirements to sustain daily FMR using the caloric value for composite PCFG prey (1.91 kJ g$^{-1}$), assuming equal proportions of the three main prey types in the region: *Holmesimysis sculpta*, *Neomysis rayii*, and *Atylus tridens* (*Hildebrand, Bernard & Torres, 2021*). A total of 20,000 Monte Carlo iterations were run per combination of covariates, and Cohen's *d* (mean, s.d) was used to compare FMR estimates across simulations (*Cohen, 1988*). We summarized Cohen's *d* values across covariate comparisons (TL, BAI, dive duration) where all other covariates

were constant (*e.g.*, we summarized Cohen's *d* across the multiple combinations where only the TL was different).

# RESULTS

## Metric exploration

### Nares area

Our nares area dataset included 183 breaths from 21 individual whales across 70 surface sequences. Seventeen individuals were observed in only one year, and four were observed in two years. There were strong relationships between the left and right measurements for nares area (slope = 0.89, Confidence Interval $(CI)_{95}$ [0.72–1.06], $R^2$: 0.81, $F_{(1,27)}$: 116.16, p: $2.78 \times 10^{-11}$), length (slope = 0.81, $CI_{95}$ [0.63–1.00], $R^2$: 0.76, $F_{(1,27)}$: 84.42, p: $8.46 \times 10^{-10}$), and width (slope = 0.70, $CI_{95}$ [0.56–0.84], $R^2$: 0.79, $F_{(1,27)}$: 100.07, p: $8.46 \times 10^{-10}$) (Fig. S3). There were positive relationships between TL and mean nares area (slope = 0.0003, $CI_{95}$ [0.0002–0.0004], $R^2$: 0.55, $F_{(1,19)}$: 23.64, p: 0.0001), length (slope = 0.01, $(CI_{95:}$ [0.01–0.02]), $R^2$: 0.46, $F_{(1,19)}$: 17.23, p: 0.0005), and width (slope = 0.01, $(CI_{95:}$ [0–0.01]), $R^2$: 0.36, $F_{(1,19)}$: 10.52, p: 0.004) (Fig. S4). The LMM comparing standardized nares area by behavior states and breath types while accounting for individual ID found no significant contrasts and had a conditional $R^2$ of 0.42 and a marginal $R^2$ of 0.01 indicating that there were no notable differences between breath types (initial, middle, terminal) for either absolute nares area ($m^2$) or length standardized nares area (Fig. S5).

### Correlation of metrics

After filtering clips, we had 557 surface sequences from 88 unique individual whales. However, not all sequences had both a preceding dive and a following dive available, and some respiration metrics had more missing values than others, resulting in different sample sizes per metric (Table S7). Summaries of the metrics are available in the supplementary (Table S6). Of the metrics, total inhalation duration and breath count were the most correlated (0.96), while the others had Pearson correlation coefficients below 0.6 (Fig. S6).

## Respiration models

### Foraging

For all models, assessment of the residual plots indicated that model assumptions were met, and prior sensitivity analysis indicated no influence of prior choice. Bayes $R^2$ values are reported in Tables 3 and 4. Results of the posterior predictive checks are available in Tables S9, S12, S13, S16.

### Recovery

*Breath count from both drone and tag data.* The results from the drone and tag data models were in agreement: TL, BAI, and preceding dive duration had a positive relationship with breath count (Fig. 2, Table 3). In the tag model, the side-swim forward tactic had a detectably higher breath count than headstanding (coefficient: 0.166, Credible Interval $(CrI_{95})$: 0.012, 0.319). There was no identifiable difference between tactics in the drone model, although subsurface stationary and the surface tactics did have the lowest predicted

**Table 3 Recovery model results.** Posterior mean and 95% credible intervals for each coefficient and the percentage of the posterior distribution that is above or below 0. Coefficients with at least 95% of the posterior distribution over (+/-) zero are bolded.

| Fixed effect | Breath count (drone) mean (95% CrI) | % over (+/-) 0 | Breath count (tag) mean (95% CrI) | % over (+/-) 0 | Total inhalation duration mean (95% CrI) | % over (+/-) 0 | Inter-breath interval (IBI) mean (95% CrI) | % over (+/-) 0 | Inhalation accumulation rate mean (95% CrI) | % over (+/-) 0 | Initial breath duration mean (95% CrI) | % over (+/-) 0 |
|---|---|---|---|---|---|---|---|---|---|---|---|---|
| Intercept | 0.83 (0.69, 0.96) | 100 | 1.05 (0.68, 1.36) | 100 | 1.01 (0.89, 1.13) | 100 | 2.64 (2.52, 2.76) | 100 | −2.24 (−2.39, −2.1) | 100 | 0.35 (0.3, 0.39) | 100 |
| Total length (TL) | 0.07 (−0.03, 0.18) | 91 | 0.08 (−0.12, 0.29) | 82 | **0.08 (−0.01, 0.17)** | **96** | **0.21 (0.11, 0.3)** | **100** | **−0.19 (−0.3, −0.08)** | **100** | 0.02 (−0.02, 0.06) | 85 |
| Body area index (BAI) | **0.11 (0.01, 0.2)** | **99** | 0.14 (−0.15, 0.42) | 87 | **0.08 (0, 0.16)** | **97** | **−0.07 (−0.15, 0.01)** | **96** | **0.09 (0, 0.19)** | **98** | −0.01 (−0.04, 0.01) | 88 |
| Bubble blast | −0.02 (−0.27, 0.23) | 55 | – | – | 0.02 (−0.22, 0.27) | 57 | 0.09 (−0.11, 0.29) | 82 | −0.08 (−0.33, 0.17) | 73 | −0.03 (−0.08, 0.03) | 84 |
| Dive duration | **0.16 (0.07, 0.23)** | **100** | **0.2 (0.15, 0.24)** | **100** | **0.16 (0.08, 0.24)** | **100** | **−0.13 (−0.2, −0.05)** | **100** | **0.13 (0.04, 0.22)** | **100** | 0.01 (−0.01, 0.03) | 79 |
| Forward swimming | −0.15 (−0.39, 0.08) | 90 | – | – | −0.09 (−0.28, 0.11) | 81 | **0.19 (0, 0.37)** | **97** | **−0.28 (−0.5, −0.07)** | **99** | −0.03 (−0.08, 0.03) | 83 |
| Subsurface stationary | **−0.39 (−0.89, 0.07)** | **95** | – | – | **−0.38 (−0.76, 0.01)** | **97** | −0.12 (−0.52, 0.29) | 72 | −0.21 (−0.73, 0.31) | 79 | **−0.09 (−0.18, 0.01)** | **97** |
| Side-swim stationary | −0.02 (−0.34, 0.29) | 54 | – | – | −0.14 (−0.41, 0.12) | 86 | −0.14 (−0.42, 0.15) | 83 | −0.02 (−0.38, 0.34) | 54 | −0.04 (−0.1, 0.03) | 88 |
| Surface tactics | **−0.44 (−0.86, −0.05)** | **99** | – | – | **−0.51 (−0.83, −0.19)** | **100** | −0.09 (−0.49, 0.3) | 67 | −0.35 (−0.88, 0.19) | 90 | **−0.12 (−0.24, −0.01)** | **98** |
| Side-swim fwd | – | – | **0.17 (0.01, 0.32)** | **98** | – | – | – | – | – | – | – | – |
| Benthic dig | – | – | 0 (−0.12, 0.12) | 52 | – | – | – | – | – | – | – | – |
| Individual random effect | 0.08 (0.01, 0.21) | 100 | 0.39 (0.17, 0.87) | 100 | 0.09 (0.01, 0.22) | 100 | 0.22 (0.09, 0.34) | 100 | 0.2 (0.05, 0.35) | 100 | 0.13 (0.1, 0.17) | 100 |
| Bayes $R^2$ | 0.41 (0.35, 0.46) | – | 0.33 (0.30, 0.36) | – | 0.26 (0.17, 0.34) | – | 0.49 (0.35, 0.59) | – | 0.46 (0.34, 0.57) | – | 0.59 (0.52, 0.65) | – |

**Table 4  Anticipation model results.** Posterior mean and 95% credible intervals for each coefficient and the percentage of the posterior distribution that is above or below 0. Coefficients with at least 95% of the posterior distribution over (+/-) zero are bolded.

| Fixed effect | Total inhalation duration | | Inter-breath interval (IBI) | | Inhalation accumulation rate | | Terminal breath duration | |
|---|---|---|---|---|---|---|---|---|
| | mean (95% CrI) | % over (+/-) 0 | mean (95% CrI) | % over (+/-) 0 | mean (95% CrI) | % over (+/-) 0 | mean (95% CrI) | % over (+/-) 0 |
| Intercept | 0.86 (0.72, 0.99) | 100 | 2.59 (2.44, 2.74) | 100 | −2.21 (−2.4, −2.02) | 100 | 0.34 (0.28, 0.41) | 100 |
| Total length (TL) | 0.04 (−0.06, 0.14) | 78 | **0.21 (0.09, 0.32)** | **100** | **−0.2 (−0.33, −0.06)** | **100** | −0.01 (−0.07, 0.04) | 65 |
| Body area index (BAI) | 0.05 (−0.04, 0.15) | 87 | **−0.11 (−0.21, −0.01)** | **99** | **0.18 (0.05, 0.31)** | **100** | 0.02 (−0.01, 0.05) | 91 |
| Bubble blast | 0.19 (−0.08, 0.46) | 92 | −0.01 (−0.26, 0.23) | 54 | −0.03 (−0.36, 0.31) | 57 | −0.01 (−0.08, 0.06) | 60 |
| Dive duration | −0.01 (−0.1, 0.08) | 61 | 0.01 (−0.08, 0.1) | 58 | −0.03 (−0.15, 0.09) | 70 | 0 (−0.03, 0.02) | 57 |
| Forward swimming | −0.07 (−0.28, 0.14) | 74 | **0.25 (0.03, 0.47)** | **99** | **−0.37 (−0.65, −0.1)** | **100** | −0.04 (−0.11, 0.03) | 88 |
| Subsurface stationary | **−0.41 (−0.83, 0)** | **97** | **1.19 (0.36, 2.02)** | **100** | **−1.42 (−2.32, −0.5)** | **100** | 0.07 (−0.11, 0.24) | 77 |
| Side-swim stationary | 0.04 (−0.28, 0.35) | 60 | 0.05 (−0.26, 0.35) | 62 | −0.17 (−0.56, 0.22) | 81 | **−0.09 (−0.18, 0)** | **98** |
| Surface tactics | **−0.48 (−0.82, −0.13)** | **100** | 0.08 (−0.42, 0.58) | 62 | −0.44 (−1.08, 0.21) | 91 | 0.17 (−0.06, 0.4) | 92 |
| Individual random effect | 0.1 (0.01, 0.25) | 100 | 0.23 (0.09, 0.37) | 100 | 0.22 (0.03, 0.42) | 100 | 0.17 (0.12, 0.22) | 100 |
| Bayes $R^2$ | 0.16 (0.08, 0.26) | – | 0.46 (0.30, 0.59) | – | 0.48 (0.35, 0.60) | – | 0.69 (0.61, 0.75) | – |

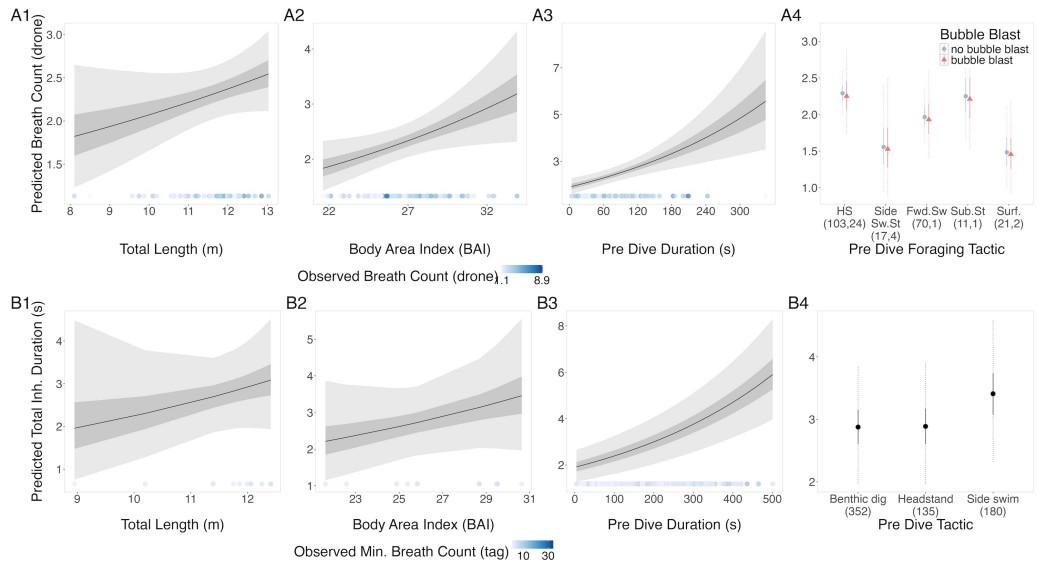

**Figure 2** **Estimated relationships between breath count per sequence from (A) drone data and (B) tag data.** Relationships between breath count per sequence and (1) total length (TL), (2) Body Area Index (BAI), (3) preceding dive duration (s) and (4) preceding dive foraging tactic and bubble blast occurrence (in A only). The response variable was transformed from the log scale back to breath counts. In columns 1, 2, and 3, the line represents the mean posterior relationship, the dark gray shaded region represents the 50% credible interval, and the light gray shaded region represents the 95% credible interval. The points along the x-axis in each plot represents the original data values, colored by the observed breath count with darker shades representing higher values. In column 4, the points represent the posterior mean breath counts, the solid lines represent the 50% credible intervals, and the dashed lines represent the 95% credible intervals. In A4 the tactics are abbreviated as follows: HS, Headstand; Side.Sw.St, Side-swim stationary; Fwd.Sw., Forward swimming tactics; Sub.St, Subsurface stationary; Surf., Surface tactics; grey circles indicate that no bubble blast occurred, while orange triangles indicated the occurrence of a bubble blast. The sample sizes per tactic are reported in parentheses under the tactic name: the first value indicates the number of observations with no bubble blast for that tactic, and the second value indicates the number of observations with a bubble blast for that tactic.

breath counts. Autocorrelation was not an issue in the residuals of either model. The dispersion parameter of the drone and tag models were 0.68 and 1.7, respectively.

*Total inhalation duration.* TL, BAI, and preceding dive duration all had a positive relationship with total inhalation duration, while bubble blast occurrence had no notable relationship (Figs. 3A1–3A4, Table 3). Side swimming stationary and the surface tactics had the shortest log-transformed total inhalation durations, specifically the surface tactics were associated with notably shorter durations compared to headstanding and the forward swimming tactics.

*Inter-breath interval (IBI).* TL had a positive relationship with mean IBI, while BAI and preceding dive duration had a negative relationship, and bubble blast occurrence had no notable relationship (Figs. 3B1–3B4, Table 3). Forward swimming was associated with the longest IBI, which was notably longer than that of side-swimming stationary.

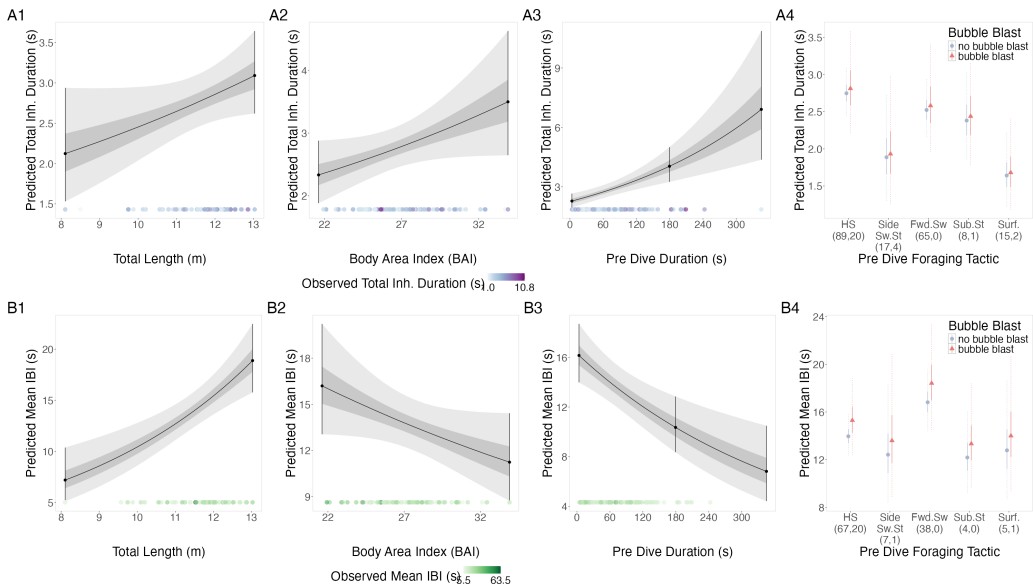

**Figure 3 Estimated relationships from the "recovery models".** Relationships between (1) total length (TL), (2) body area index (BAI), (3) preceding dive duration (s) and (4) preceding dive foraging tactic and bubble blast occurrence and (A) total inhalation duration (s) and (B) mean inter-breath interval (IBI). All response variables were transformed from the log scale to seconds. In columns 1, 2, and 3, the line represents the mean posterior relationship, the dark gray shaded region represents the 50% credible interval, and the light gray shaded region represents the 95% credible interval. Points with bars represent the mean probability and 95% credible interval at the minimum and maximum (1) TL and (2) BAI. In (3) preceding dive duration, points with bars represent the mean value and 95% credible interval at the minimum and maximum dive duration and 3- minute dive durations. The points along the *x*-axis in each plot represents the original data values, colored by the observed values of each row's respective response variable with darker shades representing higher values. In column 4, the points represent the posterior mean values, the solid lines represent the 50% credible intervals, and the dashed lines represent the 95% credible intervals. In column 4 the tactics have been abbreviated as follows: HS, Headstand; Side.Sw.St, Side-swim stationary; Fwd.Sw., Forward swimming tactics; Sub.St, Subsurface stationary; Surf., Surface tactics. Grey circles indicate that no bubble blast occurred, while orange triangles indicated the occurrence of a bubble blast. The sample sizes per tactic are reported in parentheses under the tactic name: the first value indicates the number of observations with no bubble blast for that tactic, and the second value indicates the number of observations with a bubble blast for that tactic. Note that the *Y*-axis scale is free; a fixed-scale version is available in Fig. S7.

*Inhalation accumulation rate.* TL had a negative relationship with the log-transformed inhalation accumulation rate, while BAI and preceding dive duration had a positive relationship. There was no association with bubble blast occurrence (Figs. S8A1–S8A4, Table 3). Inhalation accumulation rate was greatest after headstanding and subsurface stationary; with no overlap between the credible intervals for headstanding and the forward swimming tactics.

*Initial breath inhalation duration.* TL, BAI, preceding dive duration, and bubble blast occurrence did not have a detectable relationship with the log-transformed inhalation duration of the initial breath in a sequence (Figs. S8B1–S8B4, Table 3). Generally, side

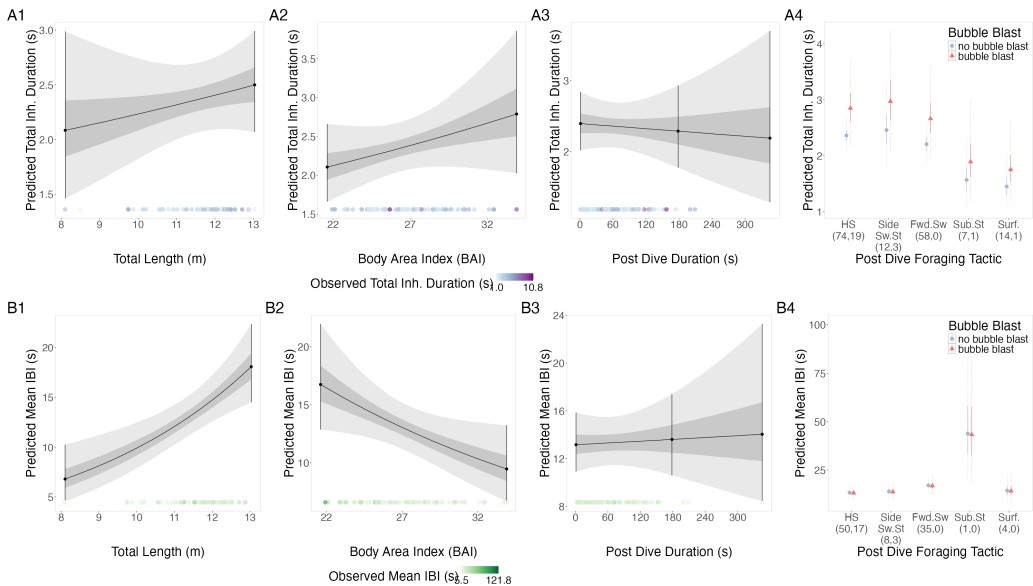

**Figure 4  Estimated relationships from the "anticipation models".** Relationships between (1) total length (TL), (2) Body Area Index (BAI), (3) following dive duration (s) and (4) following dive foraging tactic and bubble blast occurrence on (A) total inhalation duration (s) and (B) mean inter-breath interval (IBI). All response variables were transformed from the log scale to seconds. In columns 1, 2, and 3, the line represents the mean posterior relationship, the dark gray shaded region represents the 50% credible interval, and the light gray shaded region represents the 95% credible interval. Points with bars represent the mean value and 95% credible interval at the minimum and maximum (1) TL and (2) BAI. In (3) preceding dive duration, points with bars represent the mean value and 95% credible interval at the minimum and maximum dive duration and 3- minute dive durations. The points along the *x*-axis in each plot represents the original data values, colored by the observed values of each row's respective response variable with darker shades representing higher values. In column 4, the points represent the posterior mean values, the solid lines represent the 50% credible intervals, and the dashed lines represent the 95% credible intervals. In column 4 the tactics have been abbreviated as follows: HS, Headstand; Side.Sw.St, Side-swim stationary; Fwd.Sw., Forward swimming tactics; Sub.St, Subsurface stationary; Surf., Surface tactics. Grey circles indicate that no bubble blast occurred, while orange triangles indicated the occurrence of a bubble blast. The sample sizes per tactic are reported in parentheses under the tactic name: the first value indicates the number of observations with no bubble blast for that tactic, and the second value indicates the number of observations with a bubble blast for that tactic. Note that the *Y*-axis scale is free; a fixed-scale version is available in Fig. S8.

swim stationary and the surface tactics had the lowest inhalation durations, with that of the surface tactics being notably lower compared to headstanding.

### Anticipation

*Total inhalation duration.*  There were no notable relationships of the log-transformed total inhalation duration with TL, BAI, following dive duration, or bubble blast occurrence (Figs. 4A1–4A4, Table 4). However, the surface tactics were associated with a notably lower log total inhalation duration compared to other tactics.

*Inter-breath interval (IBI).*  TL had a positive relationship with log-transformed IBI, BAI had a negative relationship, while following dive duration and bubble blast occurrence

showed no relationship (Figs. 4B1–4B4, Table 4). Subsurface stationary and the forward swimming tactics were associated with the longest log IBIs.

*Inhalation accumulation rate.* Log-transformed inhalation accumulation rate had a negative relationship with TL and a positive relationship with BAI (Figs. S10A1–S10A4, Table 4). Subsurface stationary and the forward swimming tactics were both associated with low log rates compared to the other tactics.

*Terminal breath inhalation duration.* TL, BAI, following dive duration, and bubble blast occurrence did not have detectable relationships with the log-transformed terminal breath inhalation duration (Figs. S10A1–S10A4, Table 4). Side-swim stationary was associated with the lowest log terminal inhalation duration. The posterior distribution of the random effect did not overlap with 0 for 12 of the 44 individuals.

### Recovery and anticipation model
Preceding dive duration had a positive relationship with log-transformed total inhalation duration (coefficient: 0.182, $CrI_{95}$: 0.081, 0.283), while TL, BAI, and the following dive duration did not (Table S14), as suggested by the overlap of the posterior credible intervals with 0. Assessment of the residual plots indicated that model assumptions were met. The Bayes $R^2$ was 0.342 ($CrI_{95}$: 0.225, 0.440).

## Travel models
TL, BAI, and swim speed did not show a notable relationship with log-transformed respiration rate during travel, as suggested by the overlap of their 95% CrI with 0 (Fig. S11, Table S17). Specifically, there was an 84% estimated probability that respiration rate during travel increased with TL (coefficient: 0.115, $CrI_{95}$: −0.116, 0.342). BAI had a 78% estimated probability of a positive relationship (coefficient: 0.084, $CrI_{95}$: −0.131, 0.298), while there was a 73% probability of a negative relationship with swim speed (coefficient: −0.060, $CrI_{95}$: −0.259, 0.139). Assessment of the residual plots indicated that model assumptions were met, and the Bayes $R^2$ value was 0.355 ($CrI_{95}$: 0.057, 0.658).

## FMR and prey requirement simulations
Daily FMR increased with TL and BAI, decreased with dive duration, and was marginally higher for headstanding than the forward swimming tactics (Fig. 5, Table 5). The highest simulated daily FMR was for a 12-m whale with a BAI of 32 that only foraged using 60-second headstanding dives (FMR: 1,725.24 MJ day$^{-1}$, $CI_{95}$: 914.06, 2,585.37), resulting in an estimated daily prey requirement of 0.85 metric tons ($CI_{95}$: 0.45, 1.28). The lowest simulated daily FMR was for a 9-m whale with a BAI of 22 that only foraged using 300-second forward swimming dives (FMR: 719.81 MJ day$^{-1}$, $CI_{95}$: 383.49, 1,071.52), requiring an estimated 0.36 ($CI_{95}$: 0.19, 0.53) metric tons of prey per day. Controlling for all other variables, the difference in FMR given a 12- *versus* a 9-m TL had the largest Cohen's *d* (mean: 2.26, s.d: 0.07), corresponding to differences in FMR and prey requirements of 669.51 MJ day$^{-1}$ (s.d: 37.14) and 0.35 metric tons (s.d: 0.02), respectively (Table 5).

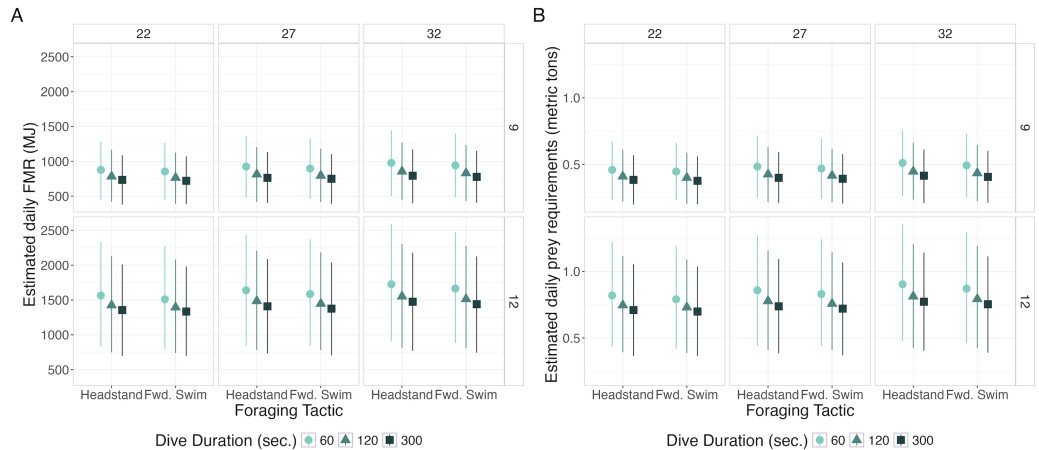

**Figure 5   Estimated daily Field Metabolic Rate (FMR; MJ) (left) and prey requirements (metric tons) (right) from simulation exercise.** Foraging tactic is on the *x*-axis, plots are faceted by body area index (BAI) in columns and total length (TL) in rows, point shapes and colors represent foraging dive durations. Each point represents the mean of the simulated values, and the bars represent 95% of the distribution.

## DISCUSSION

We found that the duration of preceding dives is more strongly related with the respiration of gray whales than the duration of following dives. The relationships between the respiration metrics and total length, body condition, and preceding dive duration were consistent in direction, while the foraging tactic effects were variable. Additionally, we found no notable relationship between swim speed and respiration rate during travel. Combined, these results provide valuable insights into the factors potentially influencing PCFG gray whale energetics in this shallow foraging niche, laying the foundation for future studies to examine the energetic consequences of disturbance and environmental change. Our work also illustrates the complementarity of drone and accelerometry tag data when used in tandem to capture different perspectives on animal behavior.

Our finding that respiration by foraging PCFG gray whales appears to be more strongly related with recovery from the previous dive rather than anticipation for the next dive provides an interesting contrast to respiration patterns of other cetaceans. The breathing rate of long-finned pilot whales (*Globicephala melas*) only correlated with the duration of the preceding dive, not the following dive (*Isojunno et al., 2018*), whereas total inhalation duration of humpback (*Megaptera novaeangliae*) and minke (*Balaenoptera acutorostrata*) whales only correlated to the following dive, not the preceding (*Nazario et al., 2022*). While we would expect similar respiration patterns among baleen whales, it is important to note that, in our study system, gray whales perform shallow dives (up to depth of ∼20 m) in contrast to the studies of other baleen whales (mean depth ∼45 m, max ∼250 m; *Nazario et al., 2022*). Furthermore, we did not find any evidence of pre-dive hyperventilation, which has been described in humpback whales (*Dolphin, 1987b*) and bottlenose dolphins (*Tursiops truncatus*; *Ridgway, Sconce & Kanwisher, 1969*). Thus, our results indicate that

PCFG gray whales may not need to physiologically anticipate dives in this shallow foraging habitat.

We found an overarching relationship between total length (TL) and various respiration metrics of gray whales. As respiration serves as a proxy for metabolic rate (*Fahlman et al., 2016*) we interpret these patterns as the result of the combined positive correlation between basal metabolic rate (BMR; the baseline energetic requirements of an individual) and size (*Kleiber, 1975*; *He et al., 2023*) and increased costs of locomotion for a large animal in shallow habitat (*Bird et al., 2024b*). The positive relationship between TL and total inhalation duration indicates that tidal volume and oxygen consumption increase with TL, aligning with previous findings (*Kooyman, 1973*; *Wahrenbrock et al., 1974*; *Fahlman et al., 2018*; *Cauture et al., 2019*; *Sumich et al., 2023*). The positive relationship between TL and inter-breath interval (IBI) equates to a negative relationship between total length and respiration rate, and consequently heart rate (*Blawas et al., 2021*). Similar relationships have been documented in several cetacean species including gray whales (*Ponganis & Kooyman, 1999*) and odontocetes (*Blawas et al., 2021*; *Spina et al., 2024*). Though the relationship between TL and respiration rate during travel was weaker (the 95% credible interval crossed zero), there was an 84% estimated probability of a positive relationship, which is contradictory to the IBI results. One possible explanation could be that whales travel close to the surface where drag is higher (*Vogel, 2020*) and the locomotive cost of stroking increases with body mass (*Williams et al., 2017*); therefore, it is possible that the positive relationship between TL and respiration rate is reflecting an increased cost of travel for larger whales. As a result of the estimated relationship between TL and the respiration metrics, and the documented positive relationship between TL and tidal volume ($V_T$, Eq. (2); *Sumich et al., 2023*), field metabolic rate (FMR) was estimated to increase with TL. However, we did not account for the energetic cost of growth for the 9 m whale, hence the difference in FMR between the 9 and 12 m simulations is likely an overestimate (*Sumich, 2021*; *Adamczak et al., 2023*).

Across all the models, higher BAI was consistently associated with an increased energetic cost. BAI had a negative relationship with IBI and a positive relationship with total inhalation duration, inhalation accumulation rate, and respiration rate. Combined, this suggests that whales in higher body condition consume more oxygen at an increased rate. Given that these whales are foraging in shallow habitat (<20 m; *Bird et al., 2024a*) and that increased body condition is linked to increased buoyancy (*Nousek-McGregor et al., 2014*; *Aoki et al., 2021*), we posit that this positive effect of BAI on energy expenditure reflects the costs of buoyancy in this shallow habitat. Furthermore, while we have never observed exhalations before diving, our results support the hypothesis that PCFG gray whales foraging in shallow habitat with higher body condition use bubble blasts to reduce the costs of buoyancy (*Bird et al., 2024b*). While deep dives are costly to marine mammals because of the duration of the breath hold (*Scholander, 1940*; *Kooyman, 1973*), several studies have found that shallow dives can be more costly, because of the cost of working against buoyant force (*Ridgway, Scronce & Kanwisher, 1969*; *Dolphin, 1987b*; *Costa, 1988*; *Costa & Gales, 2000*). The FMR simulation results showed that daily FMR was higher for whales in higher body condition, although the effect size was small.

**Table 5  Summary of field metabolic rate (FMR) and prey requirement comparisons across simulations, controlling for all other variables.**

| | Total Length (TL; m) | | Body Area Index (BAI) | | | Dive Duration (s) | | | Foraging Tactic |
|---|---|---|---|---|---|---|---|---|---|
| | 12–9 | 27–22 | 32–22 | 32–27 | 120–300 | 60–120 | 60–300 | Headstand–Fwd. Swim |
| Cohen's $d$ | 2.26 (0.07)*** | 0.17 (0.03) | 0.36 (0.04)* | 0.19 (0.02) | 0.23 (0.04)* | 0.44 (0.07)* | 0.67 (0.10)*** | 0.11 (0.03) |
| Difference in FMR (MJ/day) | 669.51 (37.14) | 47.48 (16.76) | 102.28 (35.34) | 54.79 (19.13) | 61.26 (12.06) | 124.83 (26.3) | 186.09 (37.99) | 31.11 (14.38) |
| Difference in prey requirements (metric tons) | 0.33 (0.02) | 0.02 (0.01) | 0.05 (0.02) | 0.03 (0.01) | 0.03 (0.01) | 0.07 (0.01) | 0.10 (0.02) | 0.02 (0.01) |

**Notes.**

The first row contains Cohen's d values. The next two rows contain the contrasts of FMR and prey requirements per comparison. All values are shown as mean (SD).

***Indicates a large effect size (>0.8).

**Indicates a moderate effect size (> 0.5).

*Indicates a small effect size (> 0.2).

Increased dive duration was associated with increased total inhalation duration, inhalation accumulation slope and breath count, and decreased IBI. Similar results have been found in many marine mammal species including pinnipeds (*Hindell & Lea, 1998*; *Williams et al., 2004*), odontocetes (*Isojunno et al., 2018*), and mysticetes (*Wursig, Wells & Croll, 1986*; *Dolphin, 1987b*; *Sumich, 2001*; *Keen & Qualls, 2018*). Contrastingly, the FMR simulation showed that shorter dives were more energetically expensive on a daily scale. This result may be due to the simulation design, where the simulated whale only dove using a single dive duration and kept diving until the total foraging time for the day was filled. Therefore, a simulated whale using 60 s dives dove more during one day than a whale using 300 s dives. However, recovery time did not increase with a 1:1 ratio. Rather, recovering from a single 300 s dive required less time than the total time needed to recover from five 60 s dives. As a result, over time it may be more energetically efficient to perform fewer long dives than many short dives. Such longer dives could also be more beneficial if they result in more intense bradycardia (*Thompson & Fedak, 1993*). While we chose to control dive duration in the simulations to compare our explanatory variables, future efforts should simulate a realistic mix of dive durations to more accurately estimate the daily FMR of foraging gray whales. More realistic estimates of FMR should also account for a potential diel bias in the activity budget from *Colson et al. (2024)*, which is based on tag data collected primarily during daytime. Further, as our FMR simulations were specifically designed to compare fixed effects on a daily scale, more realistic estimates of FMR that incorporate more variability in gray whale physiology and behavior would enable meaningful comparison with FMR estimates for other species.

The relationships between the different foraging tactics and the respiration metrics were variable. Headstanding and the forward swimming tactics appear to be more expensive as they had similarly high total inhalation durations, particularly in contrast to the surface tactics. The surface tactics occur just below the surface; therefore, the whale can easily surface as it is feeding. In the tag-based breath count model, the forward swimming tactics had a notably higher breath count than headstands and benthic digs, which is a surprising result considering *Colson (2023)* found that stroke rate is significantly higher during headstands. The FMR simulation results, based on the results of the respiration models, showed no notable difference between the headstanding and forward swimming tactics. Interestingly, *Colson (2023)* and *Bird et al. (2024a)* both found that headstanding and the forward swimming tactics are the most common tactics performed by PCFG gray whales and are relatively costly compared to the other tactics, suggesting that whales are able to capture sufficient prey to make using these tactics profitable (*Emlen, 1966*). Therefore, while the ontogenetic shift from forward swimming to headstanding (*Bird et al., 2024a*) may not be driven by different energetic costs of the tactics themselves, there may be increased costs associated with increased TL; perhaps headstanding provides longer individuals continued access to more profitable prey that also sustains the increased costs associated with length. Furthermore, headstanding may require time for learning and muscle development associated with growth (*Bird et al., 2024a*). Headstanding dives tend to be longer than forward swimming dives (*Bird et al., 2024b*), so headstanding may be more expensive *via* its relationship with dive duration, but our sample size was not

sufficient to explore this interaction. Future studies investigating how the interactions between morphology, behavior, stroke rate, and dive duration affect energetics will further clarify this hypothesis.

Bubble blasts had no notable effects on the respiration metrics (the 95% credible interval crossed 0). This result was not anticipated as bubble blasts are hypothesized to reduce the energetic cost of working against buoyant forces (*Bird et al., 2024b*). However, bubble blasts increase dive duration by ∼20 s during the stationary tactics (headstands and side-swim stationary; *Bird et al., 2024b*). Therefore, the effect of bubble blasts may manifest through its effect on dive duration rather than on respiration. Furthermore, the lack of detectable relationships could indicate that the bubble blast itself does not cost the whale additional energy due to the exhalation. Bubble blasts typically occur 20-30 s following the terminal inhalation, perhaps after oxygen extraction, meaning the whale does not need to inhale additional air for a bubble blast (*Sumich, 2001*).

Regarding the travel state, similar to *Mallonee (1991)* and *Stelle, Megill & Kinzel (2008)*, we found that IBIs during travel were longer than those during foraging, indicating that travel is less energetically costly than foraging. Furthermore, building on the work of *Sumich (1983)*, we expected to find a positive relationship between swim speed and respiration rate. However, we found no notable relationship between swim speed and respiration rate in either the drone or tag-based models. A factor to consider is that the whales in this study were travelling through a foraging ground, while *Sumich (1983)* observed whales on the south-bound migration. Therefore, there may be extra energetic costs of travel on a foraging ground that are unaccounted for here, such as digestion (*Williams et al., 2004*). Our sample size was also relatively small for travelling whales, as drone focal follows are limited to <15 min. A greater sample size could better inform our understanding of the cost of travel during the foraging season.

The limited flight time of the drone is a primary limitation of this study as it reduced the number of complete surface sequences we were able to observe. Limited visibility from the drone, either when the whale dove too deep or the water visibility was poor, also challenged the pilot's ability to follow the whale. However, the comparison of the drone and tag data showed that the drone data were not biased by beginning a focal follow at a random time within a foraging bout; there was no autocorrelation in the residuals of the tag breath count model, indicating that respiration between dives is independent. Our observed breath counts from drones (mean: 2.34, s.d: 1.95) and tags (mean: 3.42, s.d: 2.54) are similar to those reported for foraging gray whales off of British Columbia, Canada (mean: 2.47, s.d: 1.16; *Stelle, Megill & Kinzel, 2008*), St. Lawrence Island in the Bering Sea (mean: 4.27, s.d: 2.67; *Wursig, Wells & Croll, 1986*), and northern California, USA (mean: 3.52, s.d: 1.90; *Mallonee, 1991*).

We have shown here the power of pairing drone and tag data. The drone provided high resolution respiration metrics and high individual replicates across seven years, while the tag data provided better temporal coverage over a limited range of hours and individuals. Furthermore, the slight under- and over-dispersion of the respective drone and tag breath-count models provide an interesting contrast of the two methods. The under-dispersion of the drone model may reflect the limited observation duration, which

could result in an inflated proportion of shorter sequences. In contrast, the over-dispersion of the tag model may reflect the ability of the method to record the rarer occurrence of long sequences, with large breath counts, but on a limited number of individuals. By using these tools in tandem, we can continue to improve our understanding of respiration and energetics in free swimming large whales that are inaccessible using traditional methods.

We quantified respiration patterns using an established metric, inter-breath interval (IBI; *Wursig, Wells & Croll, 1986*; *Dolphin, 1987a*; *Stelle, Megill & Kinzel, 2008*) and novel metrics: total inhalation duration and inhalation accumulation rate. Our derived total inhalation duration metric serves as an improvement on breath count that incorporates breath-by-breath variability and estimates the total tidal volume of oxygen acquired during a surface sequence. The rate of inhalation accumulation metric integrates both total inhalation duration and IBI to reflect changes in both tidal volume and respiration rate, similar to minute volume (*Kooyman, 1973*).

There were no clear relationships between either the initial or terminal breaths with body size, condition, or dive duration. This result suggests that there is no recovery from or anticipation for a dive on the single breath scale in terms of tidal volume. The initial and terminal breath duration models had some of the highest individual-level random effect values, suggesting that some other individual level attribute might help explain this variability. We similarly found no difference in nares area by breath type, in contrast to *Nazario et al. (2022)*'s findings correlating the terminal breath area and duration to the following dive duration. Therefore, like the variability in individual breath durations, there is also unexplained variability in the nares area that needs to be further explored.

## CONCLUSIONS

In conclusion, we found that TL had a large positive effect on energetic expenditure at the dive and daily scales and dive duration had a large positive effect at the dive scale but negative effect at the daily scale. BAI had a moderate positive effect on energetic costs at the dive and daily scales, and tactic use had a negligible effect. With these findings, we provide a foundation for future studies to estimate the energetic consequences of environmental change or anthropogenic disturbance with reduced uncertainty by using breath-by-breath metrics and accounting for individual morphology, body condition, and behavior. PCFG gray whales forage in near shore habitat where they are frequently exposed to vessel traffic, noise, and changing environmental conditions (*Hildebrand et al., 2022*; *Hildebrand et al., 2024*). While behavioral and physiological responses to these stressors have been documented (*Lemos et al., 2022*; *Pirotta et al., 2023*; *Pirotta et al., 2024*), the energetic consequences have not been quantified to date, which are needed to fully assess the population level impacts of these anthropogenetic activities. Our results contribute to our understanding of respiration and energetics in baleen whales and highlight the importance of studying the energetic consequences of shallow foraging. Given that the costs associated with buoyancy are highest in large whales, these results suggest that PCFG gray whales may be smaller and skinner than their ENP counterparts (*Torres et al., 2022*; *Bierlich et al., 2023*) because of an upper limit on size after which shallow diving becomes

too energetically expensive to be beneficial. If body condition is limited by buoyancy costs in this habitat, it could lead to decreased calving rates if females cannot build sufficient reserves to sustain pregnancy, ultimately leading to population-level effects (*Blueweiss et al., 1978*; *Stewart et al., 2022*; *Pirotta et al., 2025*). However, it is worth noting that here we focus on short-term foraging costs, but the impact on an individual's annual energy budget will be dependent on foraging success and rates of energy accumulation across longer time frames, which may be differentially affected by body size. Close and long-term monitoring of PCFG gray whale population dynamics, including vital rates, is needed to examine this hypothesis.

While the PCFG represents a rare example of a baleen whale primarily foraging in shallow habitat, the dynamic relationship between body condition and FMR suggests that the energetic cost of foraging should be estimated at multiple time points in a foraging season. Furthermore, these buoyancy related costs likely affect other species as well. Shallow prey patches have been hypothesized to be profitable for foraging cetaceans due to the shorter travel time required to reach them (*Doniol-Valcroze et al., 2011*). However, *Nichols et al. (2022)* found a decrease in number of humpback whale shallow foraging dives throughout the foraging season and several studies have documented a cessation of foraging at night when zooplankton prey vertically migrate to the surface (*Goldbogen et al., 2011*; *Burrows et al., 2016*; *Nickels, Sala & Ohman, 2019*). These studies hypothesized that this lack of shallow foraging was due to decreased density of prey near the surface; however, it is possible that the cost of shallow dives due to buoyancy also decreased the profitability of the prey. Consequently, our results suggest that the central place forager framework for baleen whales should be adjusted to incorporate the increased energetic expense of shallow diving as related to an individual's size and body condition. At shallow depths, assessment of an individual's buoyancy and energetic costs may elucidate variation in foraging decisions that cannot be explained by prey density alone.

## ACKNOWLEDGEMENTS

We thank Todd Chandler, Hunter Warwick, Ines Hildebrand, Noah Goodwin-Rice, Kathryne Macallan, Ally Kane, Jen-Hsiu Ko, Ryan Giannelli, Abby Coffey, Wally Fiori, and Hali Peterson for assistance with data collection and processing. We thank Alyssa Thibodeau for providing guidance during analysis. We thank Mauricio Cantor and Tiffany Garcia for feedback on preliminary results.

### Funding

Data collection was supported by the NOAA National Marine Fisheries Service Office of Science and Technology Ocean Acoustics Program [2016 and 2017; 50-27], the Oregon Sea Grant Program Development funds [2018; RECO-40-PD], the Oregon State University Marine Mammal Institute [2019], and the Office of Naval Research Marine Mammals and Biology program [2020-2022; #N00014-20-1-2760]. The funders had no role in study design, data collection and analysis, decision to publish, or preparation of the manuscript.

## Grant Disclosures

The following grant information was disclosed by the authors:

The NOAA National Marine Fisheries Service Office of Science and Technology Ocean Acoustics Program: 2016 and 2017; 50-27.

The Oregon Sea Grant Program Development funds: 2018; RECO-40-PD.

The Oregon State University Marine Mammal Institute [2019].

The Office of Naval Research Marine Mammals and Biology program: 2020-2022; # N00014-20-1-2760.

## Competing Interests

The authors declare there are no competing interests.

## Author Contributions

- Clara N. Bird conceived and designed the experiments, performed the experiments, analyzed the data, prepared figures and/or tables, authored or reviewed drafts of the article, and approved the final draft.
- Enrico Pirotta analyzed the data, authored or reviewed drafts of the article, and approved the final draft.
- Leslie New analyzed the data, authored or reviewed drafts of the article, and approved the final draft.
- Jamie M. Cornelius analyzed the data, authored or reviewed drafts of the article, and approved the final draft.
- James L. Sumich analyzed the data, authored or reviewed drafts of the article, and approved the final draft.
- Kate M. Colson performed the experiments, authored or reviewed drafts of the article, and approved the final draft.
- K.C. Bierlich performed the experiments, authored or reviewed drafts of the article, and approved the final draft.
- Lisa Hildebrand performed the experiments, authored or reviewed drafts of the article, and approved the final draft.
- Alejandro Apolo Fernández Ajó performed the experiments, authored or reviewed drafts of the article, and approved the final draft.
- Annie Doron performed the experiments, authored or reviewed drafts of the article, and approved the final draft.
- Leigh G. Torres conceived and designed the experiments, authored or reviewed drafts of the article, and approved the final draft.

## Animal Ethics

The following information was supplied relating to ethical approvals (*i.e.*, approving body and any reference numbers):

Suction cup tagging was conducted under the IACUC approved for the Cascadia Research Collective.

## Field Study Permissions

The following information was supplied relating to field study approvals (*i.e.*, approving body and any reference numbers):

Research was conducted under NOAA/NMFS permits 16011 and 21678. As the drone was in nadir and filming whales in the open ocean, humans were never recorded nor at risk of being recorded.

## Data Availability

The data is available at figshare: Bird, Clara (2025). Data and code for ''Size and body condition drive the cost of foraging for a baleen whale in a shallow habitat''. figshare. Journal contribution. https://doi.org/10.6084/m9.figshare.26999995.v1

## Supplemental Information

Supplemental information for this article can be found online at http://dx.doi.org/10.7717/peerj.20247#supplemental-information.

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
