# Peer review of "Size and body condition drive the energetic cost of a baleen whale foraging in shallow habitat"

_PeerJ, doi:10.7717/peerj.20247_

## Round 0.1 · original submission · Major Revisions

· Academic Editor

Major Revisions

Both reviewers appreciate the work, and also highlight a number of areas where the work may be improved. Adding the suggested text and making changes to the figures will help with clarity and presentation of results.

·

Basic reporting

The overall reporting in this article is very good. The manuscript is well-written and does an excellent job contextualizing its findings in the broader research literature. I have a few comments where basic reporting should be improved.

(1) Some of the figures are overly complex and only display model predictions to the exclusion of raw and/or summarized data. Figures 3 and 4, for example, each include 16 panels and it is very difficult to make out the patterns without zooming in substantially. And since they only show model predictions, it is impossible to visually assess the models' fits to the data. I think figures with fewer, data-richer panels would be important for readers to properly interpret the results.

(2) Body size scaling figures prominently in the analysis, but the authors' terminology is imprecise at times and a source of confusion. For example, in the third paragraph of the discussion (L563-584) the authors use positive correlations and positive scaling interchangeably. However, a negatively allometric relationship (such as metabolic rate w.r.t. body size) can have a scaling exponent <1, which leads to positive correlations but *not* positive scaling.

(3) I very much appreciate the authors' attention to detail in archiving both data and code. They made it easy for me to quickly review these components of the manuscript. Most of it looks perfectly robust, but I encountered a few issues attempting to reproduce some of the analysis. Some of these are more or less critical. At the less critical end, there are some incomplete file paths (e.g., source("functions.R") should be source("scripts/functions.R")) that are easily fixed. More concerningly, I was unable to run the Stan model code as written. I spent about half an hour trying to debug it and wasn't able to find a solution. Some of these were due to mismatches between R and Stan types (e.g., subyr is a factor in R and an integer array in Stan), but others had deeper issues. I strongly recommend the authors both specify the package versions used for model fitting (some issues may be due to differing versions of rstan, for example) and verify that their code runs as written on someone else's computer.

(3a) I think reasonable people can disagree on this point, but I would argue the authors have not share the relevant raw data. The CSV files included on figshare include intermediate outputs. However, the raw tag and drone data used to generate those outputs are not there. This makes future work based on these data largely infeasible and is inconsistent with the editorial policies of PeerJ. I do acknowledge it is common for biologging data of this nature (video, accelerometer, depth, etc, as opposed to GPS-only) to only be shared in summary form like this. But it does not, in my opinion, conform with the intentions of PeerJ's data sharing policy.

Once these issues have been addressed I think this manuscript will be an excellent contribution to PeerJ.

Experimental design

This manuscript investigates diving behavior and respiratory traits of a highly unusual group of mysticetes - the Pacific Coast Feeding Group of gray whales. Among cetaceans, diving behavior and respiration is much better understood for deeper-diving balaenopterid whales with (often) more limited foraging behavior repertoires. The combination of a complex suite of behaviors with shallow diving tendencies presents a compelling edge-case for investigating the relationships between kinematics, breath holds, and foraging efficiency.

The authors have assembled an impressive dataset pairing UAS and tag-based measurements of morphology and behavior. They thoroughly analyzed these data and used their findings to project the effects of body size and foraging behavior on daily energy budgets. Although estimating the metabolic rates of large cetaceans is always fraught with assumptions, the authors have brought an excellent dataset to bear on this question.

I have a few minor suggestions regarding the reporting of the statistical analyses. The main text often reports a mean and credible interval for parameter estimates, but the supplemental tables report a mean and standard deviation. I think it'd be more helpful if that was consistent across both, and I think the mean and credible interval is the more helpful option here. Also, for the ease of the reader, please state your model formulations all in one place. They're described in the text in a few different places, but a single table that specifies, for each model, the independent and dependent variables (and their transformations, as necessary), random effects (where applicable), and error structures (Gaussian? Other?) would greatly aid readability.

Regarding the FMR simulations, I think the authors have done an excellent job incorporating uncertainty by using the posterior distributions instead of point estimates. However, I think the uncertainty in the tidal volume estimates from equation 2 may represent a substantial unaccounted source of variability. The data in figure 2 of Sumich et al. (2023) (from which equation 2 is derived) show considerable variability, especially at the largest values of t_ex * L^2, which are then further extrapolated to the body sizes of the animals in the present manuscript. If possible, I think the FMR simulations would be improved if that uncertainty could be incorporated. Perhaps by fitting a Bayesian regression model to the raw data of Sumich et al. (2023), and using the posterior distributions as you've done for other variables?

Validity of the findings

The main findings of this manuscript, i.e. the relative influences of body size and behavior on the respiratory patterns and metabolic rates of shallow-diving gray whales, are supported by the paper's results. The data, regression models, and simulations are all well-suited for the research question.

Reviewer 2 ·

Basic reporting

I commend the authors for their use of a multi-platform dataset to answer novel questions in such a challenging field as marine mammal energetics. Their successful effort to bring together these different data types on different timescales is admirable and this study provides an excellent model for future studies that aim to do the same. The manuscript is well-written though the methods and results are quite dense and there are a few critical considerations that should be addressed. The authors provide sufficient context, but I would like to see more references to recent work in the area of baleen whale energetics with direct comparison to the FMR results.

Generally, I think simplifying some of the writing in the results and discussion as well as condensing the figures and including raw data in figures could increase the readability and clarity of the manuscript. The language describing the models was at times difficult to follow so any way the readers could improve conciseness would be a positive change.

Experimental design

The authors provide a thorough introduction that identifies the gap filled by their study. However, the system described here provides a unique opportunity to study baleen whale energetics for multiple reasons: empirical measurements of respiration metrics from this species, individuals in shallow water that can be tracked over long periods, and a group that is relatively constrained to a limited foraging area. All of these reasons make this system well-posed for observational (i.e., drone) studies that are simply not possible on many other species/groups. I think the authors could emphasize this point more to describe the gap they are filling.

While the authors explore many important and interesting questions using respiratory measurements, I would like to see a clearer hypothesis that “umbrellas” these questions. For example, “we hypothesize that shallow diving gray whales incur increased energetic costs with increasing buoyancy of the individual but reduce energetic costs by increasing dive duration and therefore time spent in a hypometabolic state. We tested this hypothesis in four ways…” This sets up an interesting conflict – how do gray whales manage the challenge of shallow foraging given these two drivers? The authors touch on this in the abstract but I think this could be made clearer in setting up the study.

The experimental design of the study is otherwise clear and robustly described.

Validity of the findings

A point of emphasis that needs to be addressed is the effect of body size on energetic expenditure – this is an expected result as we expect that larger animals consume more energy. The authors do a good job of explaining this in the discussion, however I think the authors should be cautious in considering that increased costs to larger animals are inherently limiting. For example, because larger animals have lower mass-specific metabolic rates, they should be able to accumulate fat stores more quickly thereby perhaps reducing the amount of time that they must spend on the foraging ground. Thus, while their costs of foraging are higher on a daily scale this could be modulated on a seasonal scale such that perhaps they incur overall lower costs than smaller animals… while we can’t say either way whether that is true, the argument that body size may be limiting in this foraging context needs more evidence. The authors should demonstrate how these results change or stay the same when the energetic metrics are normalized by body size (whether total length or an estimated body size).

One of the main points of the manuscript is that increased buoyancy increases the energetic cost of gray whales. Can the authors comment on whether they observe exhalations before diving (or during like for bubble blasts) and, if not, can you hypothesize why gray whales would not use this to offset the costs of buoyancy?

The data in Colson et al. is biased towards day hours, as is common for short-term suction cup attached tags. I would like to see more discussion of how this might bias the FMR estimates the authors described here (i.e., are these biased high because time spent foraging may be overestimated?).

It would be useful for the authors to add a paragraph that discusses the multiple components of FMR in their interpretations of the results. As FMR includes the combined costs of BMR, heat increment of feeding/SDA, and cost of locomotion describing what the likely changes are the underpin the increase in FMR for bigger animals and longer dives seems to be missing. For example, do you think that the FMR increase for bigger animals is just BMR or is it that they have a higher cost of locomotion because the cost of being big and traveling near the surface is higher (which seems like you do per the discussion)? Similarly, for the decreased cost of long dives on the FMR simulation could this be because of reduced metabolism during dives due to the dive response?

In the conclusion the authors state “dive duration had the largest effects on daily simulated FMR.” Are you suggesting the effect of dive duration is positive (i.e., longer dives are more expensive per the resp data) or negative (i.e., long dives are cheaper per the simulation). I think this has to be clarified and I would suggest that the authors consider that both could be right? Increased resp behavior following a longer dive could potentially equate to lower resp behavior on the surface interval + dive interval scale? Longer dives would also suggest more time in hypometabolic state which would agree with the FMR simulation.

Additionally, I would like to see the authors contextualize their FMR measurements with those available for other similarly sized species… for example 719 - 1,725 MJ per day compared to ~2000 MJ per day for humpbacks on foraging grounds (Videsen et al., 2023). Based on body size is this what you would expect, higher, lower?

For the models, can the authors clarify the trends in the posterior p-value means and SDs? It seems like the posterior p-value SDs are always higher than the means indicating that the modeled data has higher variability than the actual data. Have the authors considered using different priors to attempt to address this issue? Otherwise, can they clarify whether this is an issue for the model? I think it would also be beneficial for the authors to explain why they chose to use a Bayesian framework over a frequentist one.

The full models (i.e., those typed into R) should be explicitly shown either in the main text or the supplement.

I would suggest the authors focus the main text figures on the results that they emphasize in the discussion. I don’t think the average reader needs to see the plots of every fixed effect, particularly those that do not have a biologically relevant effect (though I do think these should be retained in the supplement). If the authors do want to keep these plots, could they be combined for ease of comparison? Similarly, it would be helpful to see raw data rather than just modeled data for readers to see that the modeled trends do in fact play out in the actual data.

In addressing these points regarding the validity of the findings will be sufficient for publication.

Additional comments

The first two paragraphs of discussion start out fairly similar – can these be combined?

If the point of some of your figures is to show the difference between drone and tag as well as anticipation and recovery seeing them on the same plot would be helpful.

---

## Round 0.2 · Minor Revisions

· Academic Editor

Minor Revisions

Both reviewers thought you have addressed the majority of the changes, and recommend a minor revision.

Reviewer 1 has 2 lingering comments that I agree with:

1. Around figures, doing some more data visualizations (could be in SI material) would be helpful

2. Data sharing: like many journals, this is mandatory at PeerJ and detailed in the author instructions, https://peerj.com/about/author-instructions/ . Documenting your concerns about mis-using the raw data should be included in the data documentation / metadata.

Dryad has a 300 GB / 10 GB per file limit -- but these can be increased I think, and I'm sure there's larger file size options that I'm not aware of

·

Basic reporting

The authors have made considerable effort to respond to my earlier comments and I appreciate their revisions. The code is far more reproducible and the expanded Table 2 greatly improves interpretability.

Overall, I have two remaining issues. One I think needs to be addressed by the authors. The other is up to the discretion of the editor.

The absence of figures with raw data remains concerning. Both reviewer 2 and I raised this point. The authors added rugs to some figures, but this only shows the distribution of the data. I understand that the data are high dimensional and difficult to visualize alongside model predictions. But some visualization of the raw data is necessary for interpretation. For example, consider the breath count analysis from the drone data. The authors found strong effects from TL, BAI, and dive duration. TL and BAI were fixed per-whale, I believe. The authorrs could show the data from two or three whales (of varying sizes) with dive duration on the x-axis, breath count on the y-axis, and whale identifier in color. If foraging tactic is critical here, that could be indicated with the shape. Even if only a few whales’ data or a subset of their dives are shown, visualizing the data that way would greatly clarify the data and the models used to analyze them.

The authors disagreed with me about sharing their underlying data. As I said in my initial review, I think there are reasonable arguments on both sides of this question and it is true that the field of biologging has yet to resolve common data sharing conventions (especially for fine-resolution data, such as those collected with CATS tags as in this study). If the supporting data were accepted as currently available, this paper would fall within the current norms. However, we are in an era of shifting data sharing norms, and I think expecting authors to share more than the norm is consistent with the policies of journals and funders. I would like to respond to the author's justifications for their data sharing decisions, and I leave it to the editor to decide whether or not it meets the journal's policies.

* "First, the quantity of data (multiple terabytes) makes the data difficult to host and share online". A viable subset, for example the processed CATS data of the nine tagged whales, could be shared through FigShare, Dryad, Zenodo, or any other TRUST-compliant data repository.

* "Second, the raw data can be used for a wide variety of analyses beyond those in this manuscript, in fact several are underway, and we want to protect this work, particularly as much of it is being undertaken by graduate students." I agree that the data could be used for additional analyses, and I would argue that is the primary motivation for journals and funders to require data sharing. As for protecting graduate students' work, a 3 year embargo on the data (which should be provided by the data repository) should be sufficient. I don't believe scooping is a concern in this field, but data loss and restricted access demonstrably are.

* "Finally, we would want to engage in conversation with anyone using the raw data to ensure that certain nuances regarding the data and its collection have been properly communicated." This suggests there's a potential harm from data misuse. I don't believe those potential harms outweigh the very real harms of data loss and restriction. There have been many analyses of data made "Available upon request", all of which find high rates of data loss (Vines et al. 2014 doi:10.1016/j.cub.2013.11.014), restricted access (Tedersoo et al. 2021 doi:10.1038/s41597-021-00981-0), or biases in who the data are shared with (e.g., Acciai et al. 2023 doi:10.1038/s41597-023-02129-8). I am unaware of any studies demonstrating regular misuse of shared data.

* "We also note that we have shared similar datasets as those shared by Nazario et al. (2022), a similar paper published in PeerJ." I was a co-author on this paper, which was the outcome of lead author Emily Nazario's undergraduate thesis. Since the authors of the present manuscript are more senior and experienced, I think it's reasonable to expect a higher standard.

Experimental design

I approve of the authors' revisions in response to my initial comments. The changes look good to me, and the change they declined to implement (incorporating uncertainty in the tidal volume) is well justified.

Validity of the findings

I would make one suggestion for the conclusion. I think the emphasis on the effect of TL on energetic expenditure should be contextualized with equation 2. TL^2 is a term used to estimate FMR, so the positive effect is an assumption, rather than a finding. It may be more accurate to interpret the results as the effect of dive duration and BAI after controlling for the assumed effects of TL.

Additional comments

The code still has a few errors in it. For example, in foraging_models.R:
* Line 106: tries to save a file to a folder called "RDS_files", which doesn't exist.
* Line 114-115: variable name error, where d should be precis_nb_uas.
* Line 243: error in res_checks() [I didn't track down the ultimate source].

The code is far more reproducible than in the initial submission, and I thank the authors for their revisions. I would like to see the remaining errors ironed out.

Reviewer 2 ·

Basic reporting

Re-reading this paper I echo my kudos from the previous review and commend the authors for bringing together multiple platforms to successfully study these difficult-to-study animals. I appreciate the author’s efforts to clarify/simply the text and figures and provide a clear overarching hypothesis. Overall, I think this manuscript uses appropriate data to answer useful questions in the field of marine mammal energetics and will make a great contribution to PeerJ. That said, I have one outstanding point that should be addressed prior to publication.

The authors clarified that the FMR estimates in this manuscript are not meaningfully comparable to published FMR values for other species. This should be clearly stated in the manuscript. As an alternative, the authors could consider using a notation like cFMR (calculated FMR) or eFMR (estimated FMR) in presenting their results to emphasize that these are not directly comparable to other FMR values. Currently, it is not clear that the presented FMR values are meant to be compared to other published FMRs.

Experimental design

no comment

Validity of the findings

no comment

Additional comments

I appreciate the author’s clear commentary on each of my previous comments and have no further questions.

---

## Round 0.3 · accepted · Accept

· Academic Editor

Accept

Thanks for making the requested changes, this is a great paper!